# A Pilot Randomized Crossover Trial Assessing the Safety and Short-Term Effects of Walnut Consumption by Patients with Chronic Kidney Disease

**DOI:** 10.3390/nu12010063

**Published:** 2019-12-25

**Authors:** Pilar Sanchis, Marilisa Molina, Francisco Berga, Elena Muñoz, Regina Fortuny, Antonia Costa-Bauzá, Felix Grases, Juan Manuel Buades

**Affiliations:** 1Laboratory of Renal Lithiasis Research, University of Balearic Islands, Institute of Health Sciences Research [IUNICS-IdISBa], 07122 Palma of Mallorca, Spain; pbergamontaner@gmail.com (F.B.); antonia.costa@uib.es (A.C.-B.); 2Nephrology Department, Hospital Son Llàtzer, Institute of Health Sciences Research [IUNICS-IdISBa], 07198 Palma of Mallorca, Spain; mmolina2@hsll.es (M.M.); jbfuster@hsll.es (J.M.B.); 3ALCER Illes Balears (Association for the Fight against Kidney Diseases, Balearic Islands), 07009 Palma, Spain; elena.nutricionista@alcerib.org; 4Laboratory Department, Hospital Son Llàtzer, 07198 Palma of Mallorca, Spain; rfortuny@hsll.es

**Keywords:** walnuts, kidney disease, phytate, dietary intervention, phosphorous

## Abstract

The aim of this study of patients with chronic kidney disease (CKD) is to assess the safety of daily consumption of walnuts on the physiological levels of phosphorous, potassium, parathyroid hormone (PTH), and fibroblast growth factor 23 (FGF23), and to assess the short-term benefits of this intervention on risk factors associated with cardiovascular events. This led us to perform a prospective, randomized, crossover, pilot clinical trial examined 13 patients with CKD. Subjects were randomly assigned to a diet of 30 g of walnuts per day or the control diet. After 30 days, each group was given a 30-day washout period, and then switched to the alternate diet for 30 days. Urinary and serum levels of phosphorous and potassium, multiple vascular risk factors, and urinary inositol phosphates (InsPs) were measured at baseline and at the end of the intervention period. Our results showed that the walnut dietary supplement led to reduced blood pressure, LDL cholesterol, and albumin excretion, but had no effect on the physiological levels of phosphorous, potassium, PTH, and FGF23. This is the first report to show that daily consumption of walnuts by patients with CKD does not alter their physiological levels of phosphorous, potassium, PTH, and FGF23 when included in a sodium-, protein-, phosphate-, and potassium-controlled diet, and it could be an effective strategy for reducing cardiovascular risk in patients with CKD.

## 1. Introduction

Cardiovascular events are the most frequent cause of death in patients with chronic kidney disease (CKD), and these patients are also more likely to experience major vascular events than the general population [1]. Many epidemiological studies suggest that the consumption of nuts protects against cardiovascular disease [2,3,4,5]. A randomized study that evaluated the effects of the Mediterranean diet (which includes nuts) on the primary prevention of cardiovascular disease (PREDIMED) showed that this diet reduced the absolute risk of cardiovascular events by about 3 cases per 1000 person-years, with a relative risk reduction of 30% among high-risk people who were initially free of cardiovascular disease [6].

It is also important to consider that a high level of dietary phosphorus can exacerbate hyperparathyroidism and osteodystrophy, promote vascular calcification, and increase the risk of cardiovascular events and mortality in patients with CKD. The physiological balance of phosphate is maintained through metabolic interactions among the intestine, kidneys, and bone. Dietary restriction of phosphorus is a common strategy used in the management of patients with CKD. Thus, clinicians recommend that patients with advanced CKD adopt a diet that is low in fat (if they have dyslipidemia) and low in phosphorus and potassium (if these are elevated or if they have stage 4 CKD). This usually means restricted consumption of dairy products and foods that are rich in fiber-including nuts-because these foods are also rich in phosphorus [7,8] and potassium. Dietary protein is also an important source of phosphorus. Recent data indicated that restriction of phosphorus consumption can lead to inadequate intake of nutrients known to prevent of cardiovascular events [9,10].

Culinary nuts, including true nuts, nut-like drupes, and nut-like seeds, are very rich in fat (48 to 63 g/100 g) and typically have abundant mono-unsaturated fatty acids (MUFAs, mainly oleic acid) and polyunsaturated fatty acids (PUFAs, mainly linoleic and alpha-linolenic acids). Nuts also have high levels of fiber, typically 5% to 9%, and are important sources of arginine, potassium, vitamin E, and other bioactive compounds (myo-inositol hexaphosphate and pyridoxamine). The unique nutritional composition of nuts may partly explain their beneficial health effects reported in previous prospective cohort studies and short-term interventional studies [11].

Phytate (phytic acid or myo-inositol hexaphosphate) is natural product present in fiber-rich foods, such as whole grains, legumes, and nuts, and are one of the most effective dietary inhibitors of calcium salt crystallization [12]. In particular, urinary inositol phosphates (InsPs) inhibit the crystallization and formation of calcium stones [13,14]. Daily intake of phytate can also inhibit vascular calcification [15,16,17,18] and prevent osteoporosis [19,20]. The traditional Mediterranean diet assumes the consumption of about 1 g of phytate per day [21], an amount that leads to optimal InsPs concentrations in blood and urine and protects against calcium lithiasis, osteoporosis, and vascular calcifications. Alternative diets, in which there are low levels of fiber, lead to greatly reduced levels of InsPs. In particular, after about 15 to 20 days of a diet without phytates, individuals have almost undetectable levels of InsPs in their blood and urine. The urinary level of InsPs is associated with the consumption of phytate [12].

We have preliminary data indicating that patients with CKD and undergoing hemodialysis have undetectable levels of InsPs in their blood. This may be because of their adoption of a low-fiber diet and/or the elimination of InsPs during the dialysis process (unpublished data). However, we have no data on how the blood and urinary InsPs levels of patients change as CKD progresses from stage 1 to stage 4. Our recent study demonstrated that greater consumption of phytate (based on a food questionnaire) protected against vascular calcification in patients with CKD [22]. This suggests that adequate consumption of phytate (1 g/day) may protect against vascular calcification in these patients. Other factors also regulate calcification, such as parathyroid hormone (PTH) and vitamin D. Recent studies determined that fibroblast growth factor 23 (FGF23, from bone) and Klotho (KL, from kidney) form an endocrine network that controls urinary phosphate excretion [23,24].

PTH also plays a central role in the regulation of mineral metabolism. Thus, patients with CKD experience increased PTH levels as renal function decreases due to phosphate retention, hypocalcemia, and decreased production of vitamin D, 25(OH)D. The initial stages of CKD are characterized by a decreased ability to eliminate ingested phosphate, and this causes an increase in bone secretion of FGF23, which leads to phosphaturia, as the body attempts to maintain an appropriate level of phosphorus in the blood. All of this occurs at the expense of a decrease in vitamin D and an increase of PTH, resulting in secondary hyperparathyroidism. On the other hand, a recent study found that a diet rich in fiber is associated with decreased inflammation (C-reactive protein, CRP) and reduced overall mortality in patients with CKD; individuals without CKD who adopted a fiber-rich diet also experienced a reduction of CRP but did not experience a decreased mortality [25].

All these studies suggest that the moderate consumption of nuts by patients with CKD may prevent cardiovascular events. However, very few interventional studies have assessed the clinical effects of nut consumption by patients with CKD. Moreover, no studies have examined the association of nut consumption with physiological levels of InsPs in patients with CKD. The aim of this study of patients with CKD was to evaluate the safety of a diet rich in walnuts on the physiological levels of phosphorous, potassium, PTH, and FGF23, and the short-term effects of this diet on classic markers of cardiovascular risk.

## 2. Materials and Methods

### 2.1. Subjects and Study Design

This was a single-center, randomized, crossover, open-label study. Fifteen subjects with CKD were prospectively and consecutively enrolled from our outpatient clinic at the Nephrology Department of Hospital Son Llàtzer, a public tertiary care center that covers 250,000 residents of urban and rural areas in the Balearic Islands (Spain).

Volunteers were eligible if they were older than 40 years, had CKD stage 3 or 4, were not undergoing renal replacement therapy, and provided informed consent. CKD was diagnosed based on an estimated glomerular filtration rate (eGFR) of 15 to 59 mL/min/1.73 m^2^, calculated as previously described [26]. No patient had cancer or clinically significant cardiovascular, liver, or end-stage kidney disease. Thirteen volunteers (87%) successfully completed the study. The minimum number of patients, assuming that the percentage of patients without significant changes in serum phosphate potassium, PTH and FGF23 would be approximately 95% for both treatments, was 11 patients per group, considering a 0.3 equivalence margin, 80% power, and a 95% confidence interval.

Subjects were randomly assigned to the walnut diet or the control diet for the initial 30 days, subjected to a 30-day washout period, and then switched to the other diet for 30 days. All patients were followed for an additional 30 days (Figure 1). Patients were asked not to change their medications (dose or drug) during the study period.

An independent researcher with no clinical involvement in the trial created a randomization sequence using Excel software (Microsoft Office, 2010), with a 1:1 allocation using a random block of 4. After obtaining each patient’s consent, the clinician contacted the independent researcher who performed the recruitment process for allocation consignment, and enrolled and assessed participants in sequence. After randomization, there was no blinding of study participants or of researchers who administered the interventions, but there was blinding of researchers who assessed outcomes.

### 2.2. Dietary Intervention

Patients received a diet with walnuts or a control diet, the same diet without walnuts. The walnut supplementation was 30 g per day, and was given in the middle of the morning. However, 60 g of unsalted white bread with 5 g of olive oil was consumed in the middle morning in the control diet. Before each intervention, all subjects received oral and written information about their diet plans, which were 2000 kCal or 1650 kCal, according to the caloric requirement of each patient. The dietary plan consisted of a daily meal plan, with five meals per day. Participants were requested to follow the meal plan as much as possible, and to report any meal that differed from what was stipulated.

The diet plan for both groups included fruit, vegetables, fish, shellfish, meat, eggs, olive oil, and low-fat dairy products. This diet was elaborated by ALCER (Spanish Association for the Fight Against Kidney Diseases) and is in accordance with the dietary recommendations of Spain for people with CKD, regarding macronutrient composition, dietary fiber, minerals, and vitamins (Table 1). Mifflin St Jeor equation were used, and daily protein intake was established at 0.8 g protein/kg of weight.

During the study intervention, a dietician followed each patient weekly to ensure that the dietary plan was followed. The dietician also checked for compliance (consumption of at least 80% of walnuts) at every visit by counting the walnuts bags (30 g/bag).

### 2.3. Outcome Measures

The main outcome measures were the physiological levels of phosphorus, potassium, PTH, and FGF23. The secondary outcomes were the classic cardiovascular risk factors such as glycated hemoglobin, fasting glucose, creatinine, blood lipids, blood pressure, body mass index (BMI), C-reactive protein, alkaline phosphatase, tubular reabsorption, glomerular filter, uric acid, microalbuminuria, and protein excretion.

Clinical histories were extracted from the electronic medical records. Furthermore, laboratory analyses and physical examinations were prospectively collected during the trial. Physical and anthropometric measurements were determined by qualified personnel while the subjects were barefoot and wearing light clothes. 24 h-urine, 2 h-urine and blood samples were taken before and after the dietary interventions, and after the follow-up period. Blood samples were collected in the morning (after 12 h of fasting), allowed to stand for 30 min at room temperature, and the serum was then separated by centrifugation. Most urinary and blood determinations were performed by potentiometry, photometry, molecular absorption spectrometry or immunochemiluminescence using an automated analyzer (Architect ci16200, Abbott, Chicago, IL, USA). Hemoglobin was determined by espectrophotometry (Cell-Dyn Sapphire). The glycated hemoglobin was determined by high performance liquid chromatography (ADAMS A1C HA-8180V, Menarini, Florence, Italy). Highly sensitive CRP (hs-CRP) was measured by nephelometry (IMMAGE 8000, Beckman Coulter, Brea, CA, USA). All samples were run in duplicate, and the coefficients of intra- and inter-assay variation were below 10%.

Blood pressure was measured 3 times consecutively after 5 min of rest, while the subject was sitting quietly. The average of the second and third measurements was recorded. Patients using anti-hypertensive drugs and those with systolic blood pressure of 140 mmHg or more and/or diastolic blood pressure of 90 mmHg or more were categorized as having hypertension [27].

### 2.4. Measurement of Urinary InsPs

The urinary level of InsPs was measured at 2 h after the first urine of the morning. For this test, 20 mL of fresh urine was acidified with HCl (1:1) to pH 3, and then diluted with 20 mL of milli-Q water. This solution was transferred to a 100 mL beaker containing 0.5 g of AG1-X8 resin (anion exchange resin), without previous conditioning. This mixture was stirred with an orbital stirrer at 160 rpm for 15 min. The resin and urine were then transferred into a 20 mL solid phase extraction (SPE) tube with a frit, and urine was passed through to separate it from the resin. The resin was then washed with 120 mL of 50 mM HCl and 2 × 5 mL of deionized water. Finally, phytate (InsP6) was eluted by 4 × 1 mL of 2 M NaCl, with contact between the resin and each 1 mL portion maintained for 5 min by mixing with an orbital stirrer (180 rpm). The final 4 mL of eluate was collected into a single tube, and the solution was mixed prior to quantification. InsPs were determined by indirect InsP6 analysis of the eluate using the aluminum-pyrocatechol violet (Al-PCV) system [28]. The two reagents (RI and RII) were prepared daily. RI was a mixture of 0.6 mL of 4 mM Al(NO3)3 and 4.4 mL of 1.5 M acetic acid/acetate buffer at pH 5.2; RII was a mixture of 0.6 mL of 5.6 mM PCV and 4.4 mL of deionized water. InsP6 standards in the range 1 to 10 µM were prepared in 2 M NaCl. The assays were performed in 96-well plates, with each well containing 30 µL RI, 290 µL of a standard or eluate, and 30 µL of RII. After incubation for 15 min, absorbance was measured at 570 nm. All samples were assayed in duplicate. This method does not discriminate between InsP6 and other InsPs, so the measured parameter is reported as ‘phytic acid equivalents’.

### 2.5. Statistical Analysis

Data are presented as means (standard deviations, SDs), medians (interquartile ranges, IQRs), or numbers (percentages). Inter-group comparisons at baseline (T0, before the intervention) were analyzed using the independent-samples t-test or the Mann-Whitney U test for continuous variables, and the chi-square test or Fisher’s exact test for categorical variables. Intra-group differences (before the intervention [T0] vs. after the intervention [T1]; after the intervention [T1] vs. after follow-up [T2]) were evaluated using a paired-samples t-test or the Wilcoxon signed-rank paired test for continuous variables, and the McNemar test for dichotomized variables. Inter-group comparisons (after the intervention and after the follow-up period) were assessed using analysis of covariance and Fisher’s exact test, with adjustment for changes in categorical and continuous variables according to baseline values. Bivariate associations were evaluated by Pearson’s or Spearman’s correlation coefficient. A physiological level of phosphorus or potassium that was two-fold above or below the standard deviation before the dietary intervention was considered significant. A two-tailed p-value less than 0.05 was considered statistically significant. Statistical analyses were performed using SPSS version 23.0 (SPSS Inc., Chicago, IL, USA).

### 2.6. Ethical Considerations

The study design was approved by the Research Committee of Hospital Son Llàtzer and Research Ethics Committee of Balearic Islands (CEI-IB) (IB2426/14 PI). All patients provided written informed consent before participation. All experiments were performed in accordance with relevant guidelines and regulations.

## 3. Results

### 3.1. Baseline Characteristics of Patients

Thirteen CKD patients (7 females and 6 males) completed the clinical study (Table 2). The median patient age was 71 years [Q1, Q3: 66, 77]. One patient (7.7%) consumed alcohol, 5 patients (39%) smoked tobacco, 5 patients (39%) had diabetes type II, and 12 patients (92%) had hypertension. The median eGFR was 42 mL/min/1.73 m^2^ [Q1, Q3: 34 to 47], and analysis of CKD stage indicated that 5 patients (38%) had stage 3a, 6 patients (46%) had stage 3b, and 2 patients (15%) had stage 4. Table 2 also shows the medications used by these 13 patients.

After group allocation, analysis of the anthropometric and laboratory variables indicated the two groups had no differences in the levels of blood or urinary phosphate or potassium, nor in any of the other analyzed variables (Table 3).

### 3.2. Effect of Walnut Consumption on Classic Cardiovascular Risk Factors

Table 4 show the changes in vascular risk factors after the diet intervention. As can be seen, the walnut diet led to a significantly reduced systolic blood pressure (−4 mmHg [Q1, Q3: −28, 0] vs. 5 mmHg [Q1, Q3: −10, 13]; *p* = 0.040) and a significantly decreased level of LDL cholesterol (−5.40 mg/dL [Q1, Q3: −12.5, −3.3]; *p* = 0.016). The decline in LDL cholesterol tended to be greater than after the walnut diet (*p* = 0.077). Analysis of urinary parameters indicated the walnut diet led to significantly reduced albumin excretion (−19 mg/24 h [Q1, Q3: −174, −3.3]; *p* = 0.011), and this decrease was also greater than after the control diet (*p* = 0.029). The walnut diet also led to a significantly reduced level of urinary chlorine (−26.40 mEq/24 h [Q1, Q3: −68.00, −5.50]; *p* = 0.028). There were no significant differences in the periods or treatment-order effects.

### 3.3. Urinary Excretion of InsPs

Analysis of urinary InsP levels (Figure 2) indicated that patients had similar urinary levels before the walnut and control diet (0.23 mg/L [Q1, Q3: 0.15, 0.44] vs. 0.33 mg/L [Q1, Q3: 0.25, 042]; *p* = 0.287). After 30 days, the walnut diet led to a significant increase in urinary InsPs (0.10 mg/L [Q1, Q3: 0.04, 0.20] vs. −0.04 mg/L [Q1, Q3: −0.22, 0.07]; *p* = 0.034).

### 3.4. Safety of Walnuts and Adverse Events

Figure 3 shows the changes in blood levels of phosphorus, potassium, PTH, and FGF23 in each patient following the walnut diet and control diet. There were no differences in any of these parameters after the dietary interventions, and all levels were similar to those at baseline. Moreover, no patient experienced an absolute change in any of these parameters that was more than twice the standard deviation of the baseline value.

All patients exhibited good tolerance to the walnut diet. There were no serious adverse events (death, life-threatening events, or events placing a patient in jeopardy or leading to hospital admission) and no dropouts related to walnut supplementation.

## 4. Discussion

This is the first prospective, crossover, randomized trial of patients with CKD to examine the safety of walnut consumption on the physiological levels of phosphorous, potassium, PTH, and FGF23, and to assess the short-term cardiovascular benefits of daily walnut consumption. Our results indicated that the daily intake of 30 g of walnuts for 30 days led to no significant changes from baseline in the physiological levels of phosphorous, potassium, PTH, and FGF23. Our finding that walnut consumption did not increase the FGF23 level indicates that despite administering this P-rich food, there was no P retention, which would require significant renal activity to compensate for the excess. In addition, daily walnut consumption by CKD patients led to improvements of three factors associated with cardiovascular disease: systolic blood pressure, LDH cholesterol, and urinary albumin.

Numerous randomized studies have reported that diets enriched with nuts consistently reduced blood pressure [29,30] and LDL cholesterol [31,32]. The effects of a nut-rich diet on HDL cholesterol have varied among studies, although the relationship between total cholesterol and HDL cholesterol usually decreases [33]. Regarding urinary albumin excretion, some studies found that a diet rich in whole grains, fruit, and low-fat dairy foods was associated with lower urinary albumin excretion [34]. Microalbuminuria is associated with the development of premature cardiovascular mortality in patients with CKD, and there is a positive correlation of blood pressure with urinary albumin excretion [35]. Thus, one of our remarkable findings is that both measures significantly decreased in patients eating walnuts. It is, therefore, possible that the moderate consumption of walnuts can reduce kidney damage and inflammation and improve endothelial function, and thereby reduce urinary albumin excretion in patients with CKD.

Our results are in accordance previous studies which reported that dietary phosphorus of vegetable origin (culinary nuts), which is mainly in the form of phytate, has lower absorbance (~10%) than phosphorus of animal origin (40 to 60%) [9,10,36,37]. Dietary phosphorus can also be classified as organic (from plants and animals) or inorganic (from inorganic preservatives). Protein-rich foods of vegetable and animal origin are abundant in organic phosphorus. Up to 100% of the inorganic phosphorus (phosphate salts) in processed foods can be absorbed. This includes the phosphorus present in some processed foods, such as cheese and some soft drinks [36,37]. However, very few intervention studies have examined the effect of different sources of dietary phosphorus on phosphorus metabolism in patients with CKD [37]. One study measured phosphate levels initially and after 3 months in 145 patients who received education regarding avoidance of foods with phosphorus additives (inorganic phosphorus) and in 134 controls who continued to receive usual care. These two groups had similar serum phosphorus levels at the beginning, but the intervention group had a greater decline of serum phosphorus at 3 months [38]. Another recent study compared a diet with protein of vegetable origin and a diet with protein of animal origin in 9 patients with CKD, and found that 1 week of a vegetarian diet reduced serum phosphorus levels [39]. Our results, together with these previous data, suggest that dietary phosphorous management in patients with CKD should focus on the source of phosphorous (animal vs. plant), the chemical structure of the phosphorus (inorganic vs. organic), and the protein-phosphorus ratio.

Considering that phytate, the main source of phosphorus in culinary nuts, is less absorbable and prevents vascular calcification [15,16,17,18,22] and reduces the levels of advanced glycation end-products (AGEs) [40], we suggest that a diet with the lowest possible amount of inorganic P, a low P-to-protein ratio, and adequate consumption of culinary nuts could be safe and beneficial for patients with CKD.

The present study had a few limitations. First, we only examined 13 patients, all of whom were from a single medical center. The small sample size in this safety study resulted in limited statistical power to demonstrate the benefit of walnuts, and means that our findings may have limited generalizability. Another limitation is that the dietary intervention was not blinded. Additionally, the walnut intervention only lasted 30 days, which may have been insufficient for the full manifestation of all the effects of walnut consumption. In this sense, the statistically significant changes found in albumin excretion, blood pressure and LDL cholesterol are not very high, and their clinical relevance is low. Thus, large, multicenter, long-term, and blinded placebo-controlled studies must be performed to assess the safety and long-term benefits and risks of walnut consumption more completely in patients with CKD.

## 5. Conclusions

Our data suggest that the daily consumption of walnuts by patients with CKD had no impact on their physiological levels of phosphorous, potassium, PTH, and FGF23 when included in a sodium-, protein-, phosphate-, and potassium-controlled diet. Furthermore, daily consumption of walnuts may have provided protection against some cardiovascular risk factors. This study provides further evidence that the daily intake of nuts appears to be a safe when included in a sodium-, protein-, phosphate-, and potassium-controlled diet, and it could be an effective strategy for reducing cardiovascular risk in patients with CKD.

## Figures and Tables

**Figure 1 nutrients-12-00063-f001:**
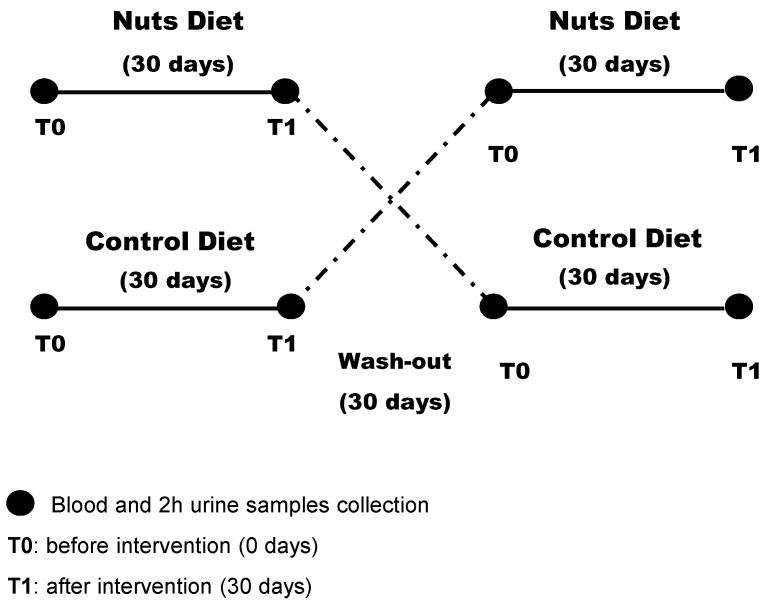
Design of the randomized crossover study of patients with chronic kidney disease (*n* = 13).

**Figure 2 nutrients-12-00063-f002:**
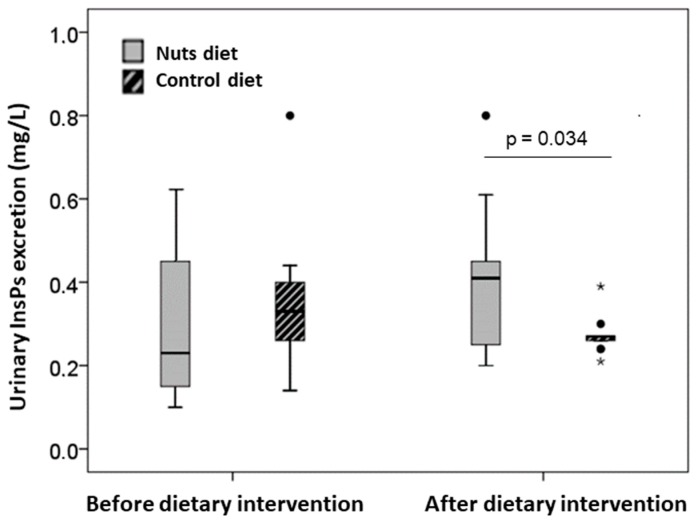
Urinary InsPs excretion levels (expressed as phytic acid equivalents, mg/L) before and after both dietary interventions. Values are expressed as median (interquartile range). Inter-group comparison after dietary intervention was performed by analysis of covariances after adjusting for baseline values. The dark lines in the middle of the boxes are the medians. The bottom and the top of the box indicates the 25th and the 75th percentiles, respectively. The T-bars that extend from the boxes are the inner fences. The points are outliers and the asterisks are extreme outliers.

**Figure 3 nutrients-12-00063-f003:**
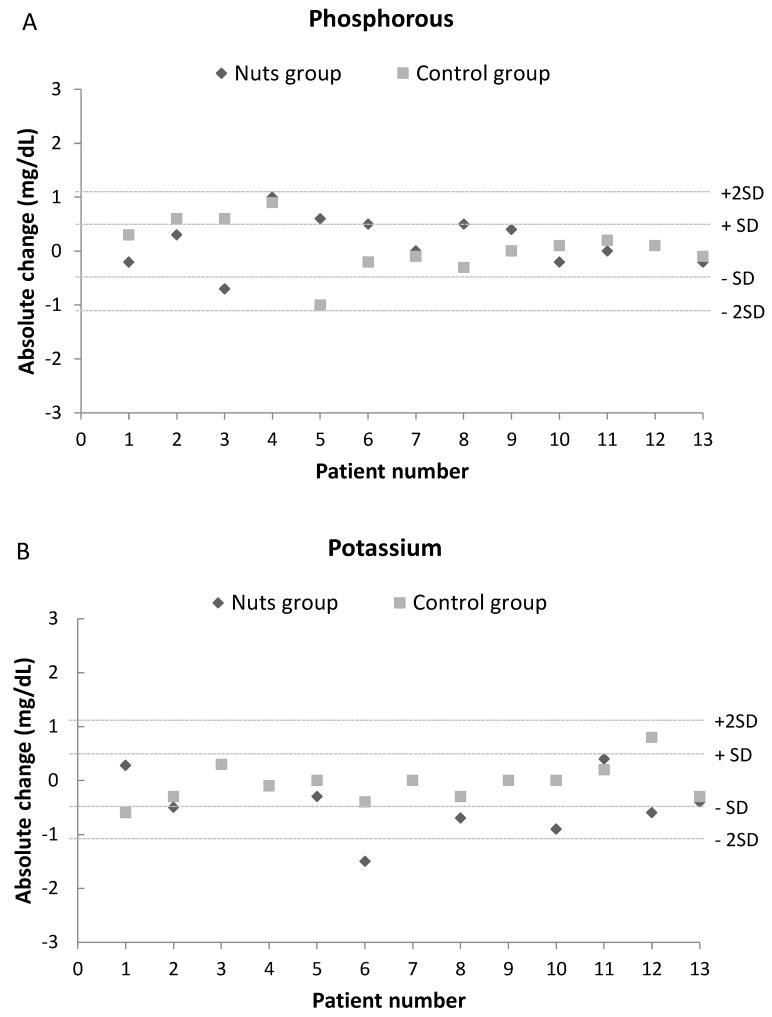
Effect of dietary intervention with nuts in blood levels of phosphorus (**A**), potassium (**B**), PTH (**C**) and FGF23 (**D**). Values are expressed as absolute change (T1-T0). A change over twice the common standard deviation of baseline values was considered a remarkable difference.

**Table 1 nutrients-12-00063-t001:** Dietary composition of both dietary interventions.

	2000 Kcal Diet with Nuts	2000 kcal Diet No Nuts	1650 kcal Diet with Nuts	1650 kcal Diet No Nuts
Protein (g/day)	65.7 (0.8 g/kg/day)	66.1 (0.8 g/kg/day)	55.5 (0.8 g/kg/day)	55.9 (0.8 g/kg/day)
Lipids (g/day)	99.1	85.7	81.4	68
Carbohydrates (g/day)	209.0	240.2	176.7	208
Fiber (g/day)	22.8	23.1	20.6	20.9
Sodium (mg/day) *	206.9	206.9	235.3	234.4
Potassium (mg/day)	2190 (0.6 mmol/kg/day)	2106 (0.6 mmol/kg/day)	2123 (0.7 mmol/kg/day)	2039 (0.7 mmol/kg/day)
Phosphorus (mg/day)	855.7	802.0	824.6	771

* although diets were made without added salt. one teaspoon of salt (5 g salt, 2000 mg sodium) a day was allowed to be added to the diet.

**Table 2 nutrients-12-00063-t002:** Baseline characteristics of patients (*n* = 13). Each value is given as median (interquartile range) or number (%).

Baseline Characteristics (*n* = 13)		
Age (Years)	71	(66–77)
Sex (female)	6	(46.2%)
Weight (kg)	88	(70–98)
Body mass index (BMI) categories		
<25 kg/m^2^	4	(30.8%)
25–30 kg/m^2^	3	(23.1%)
30–35 kg/m^2^	6	(46.2%)
Comorbidities		
Smoking (ex or yes)	5	(38.5%)
Alcohol (ex or yes)	1	(7.7%)
Diabetes	5	(38.5%)
Hypertension	12	(92.3%)
CKD Parameters		
Estimated glomerular filter (mL/min/1.73 m^2^)	42	(34–47)
Chronic Kidney Disease Stage		
3 a (eGFR 59–45 mL/min/1.73 m^2^)	5	(38.5%)
3 b (eGFR 44–30 mL/min/1.73 m^2^)	6	(46.2%)
4 (eGFR 29–15 mL/min/1.73 m^2^)	2	(15.4%)
CKD cause		
Vascular disease	5	(38.5%)
Diabetic Nephropathy	4	(30.8%)
Glomerulonephritis	5	(38.5%)
Pyelonephritis	2	(15.4%)
Polycistic disease	1	(7.7%)
Hereditary/congenital	0	(0.0%)
Systemic diseases	9	(69.2%)
Medication Use		
Angiotensin-converting enzyme inhibitors/angiotensin II receptor-blocking agents	9	(69.2%)
Beta-blockers	5	(38.5%)
Calcium antagonists	6	(46.2%)
Statins	8	(61.5%)
Fibrates	1	(7.7%)
Antiplatelets	3	(23.1%)
Oral anticoagulants	4	(30.8%)
Furosemide or triamterene	4	(30.8%)
Thiazides	1	(7.7%)
Potassium sparing diuretics	1	(7.7%)
Ion-exchange resins	0	(0.0%)

**Table 3 nutrients-12-00063-t003:** Anthropometric and laboratory values before starting each diet. Each value is given as median (interquartile range). The significance of differences between groups (inter-group comparison) was determined using the Mann-Whitney U test or an independent-samples *t*-test.

	Before Nuts Diet (To)		Before Control Diet (To)		Inter-Group *p*-Value
	Median	(Q1–Q3)	Median	(Q1–Q3)	
BMI (kg/m^2^)	27	(24–32)	32	(25–34)	0.457
Systolic Blood Pressure (mm Hg)	150	(126–162)	129	(122–146)	0.281
Diastolic blood pressure (mm Hg)	76	(71–82)	69	(66–80)	0.072
Heart rate (bpm)	62	(54–72)	66	(59–76)	0.329
Serum Parameters					
pH	7.4	(7.3–7.4)	7.4	(7.4–7.4)	0.697
pCO2 (mmHg)	52	(46–56)	46	(44–49)	0.027
Bicarbonate (mEq/L)	27	(26–30)	26	(25–27)	0.117
Base excess (mmol/L)	3.3	(1.6–5.6)	1.3	(0.4–3.3)	0.054
Phosphorous (mg/dL)	3.4	(2.9–3.7)	3.5	(3.1–4.0)	0.681
Potassium (mg/dL)	4.6	(4.2–5.1)	4.3	(4.2–4.5)	0.105
HDL cholesterol (mg/dL)	52	(41–59)	44	(38–62)	0.681
LDL cholesterol (mg/dL)	87	(73–121)	83	(70–111)	0.939
Cholesterol (mg/dL)	148	(133–197)	161	(129–201)	0.817
Triglycerides (mg/dL)	85	(73–111)	94	(78–132)	0.412
Creatinine (mg/dL)	1.5	(1.3–1.6)	1.6	(1.4–1.8)	0.555
Glucose (mg/dL)	104	(97–127)	113	(98–153)	0.608
Glycated hemoglobin (%)	5.7	(5.5–7.0)	5.8	(5.6–7.4)	0.439
Hemoglobin (g/dL)	14	(12–15)	14	(12–15)	0.980
FGF 23 (ng/mL)	115	(68–174)	112	(66–135)	0.758
Sodium (mEq/L)	140	(140–143)	140	(140–141)	0.638
Chlorine (mEq/L)	107	(105–108)	106	(105–108)	0.815
Albumin (mg/dL)	4.2	(4.1–4.5)	4.3	(4.1–4.4)	0.755
Calcium (mg/dL)	9.4	(9.2–9.8)	9.2	(8.9–9.6)	0.226
Magnesium (mg/dL)	2.1	(1.8–2.1)	1.9	(1.8–2.1)	0.340
PTHi (pg/mL)	89	(53–98)	69	(40–139)	0.758
Alkaline phosphatase (u/L)	79	(65–109)	79	(70–100)	0.959
25-OH Vitamin D3 (ng/mL)	28	(12–38)	27	(18–36)	0.918
C-reactive protein (mg/L)	2.3	(1.7–3.0)	2.1	(1.0–4.9)	0.837
Urinary Parameters					
pH	5.6	(5.2–5.8)	6.2	(6.0–6.6)	0.029
Phosphate (mg/24 h)	707	(332–838)	442	(325–893)	0.980
Potassium (mEq24 h)	57	(49–93)	59	(40–76)	0.538
Tubular reabsorption (%)	70	(70–85)	80	(76–87)	0.183
Urea (mg/dL)	58	(52–76)	55	(39–70)	0.681
Glomerular filter (mL/min/1.73 m^2^)	45	(33–49)	42	(30–48)	0.663
Microalbumin (mg/24 h)	223	(27–865)	60	(24–369)	0.479
Uric Acid (mg/24 h)	6.5	(5.3–7.1)	6.4	(4.6–7.2)	1.000
Protein (mg/24 h)	591	(125–1369)	287	(127–718)	0.579
Oxalate (mg/24 h)	24	(19–27)	27	(19–34)	0.248
Sodium (mEq/24 h)	128	(96–193)	132	(116–191)	0.837
Chlorine (mEq/24 h)	127	(98–201)	134	(103–189)	0.918

**Table 4 nutrients-12-00063-t004:** Changes in vascular risk factors after the diet intervention (T1). Un-adjusted within groups changes (before, T0 vs. after intervention, (T1) are given as median (interquartile range). Intra-group analysis (T0 vs. T1) used a paired-sample Wilcoxon signed-rank test or paired-samples *t*-test to determine the significance of differences. Inter-group analysis (Nuts diet vs. control diet) used analysis of covariances after adjusting for baseline levels to determine the significance of differences.

	After Nuts Diet (T1)			After Control Diet (T1)			InterG *p*-Value
	Median	(Q1–Q3)	intraG *p*-Value	Median	(Q1–Q3)	intraG *p*-Value	
BMI (kg/m^2^)	0.0	(−0.6–0.3)	0.919	−0.2	(−0.7–0.8)	0.656	0.719
Systolic BP (mm Hg)	−4.0	(−28.0–0.0)	0.021	5.0	(−10.0–13.0)	0.624	0.040
Diastolic BP (mm Hg)	−4.0	(−14.5–1.0)	0.054	−3.0	(−9.0–5.0)	0.581	0.342
Heart rate (bpm)	2.0	(−3.5–6.5)	0.307	−2.0	(−9.0–0.5)	0.099	0.068
Serum Parameters							
pH	−0.02	(−0.03–0.02)	0.356	−0.01	(−0.03–0.00)	0.237	0.815
pCO2 (mmHg)	0.0	(−4.4–3.0)	0.814	1.0	(−1.0–5.0)	0.099	0.269
Bicarbonate (mEq/L)	0.5	(−1.5–1.1)	0.944	0.70	(−2.2–2.2)	0.600	0.590
Base excess (mmol/L)	0.3	(−1.2–1.2)	0.753	0.50	(−0.8–3.6)	0.196	0.355
Phosphorous (mg/dL)	0.1	(−0.2–0.5)	0.212	0.10	(−0.2–0.5)	0.478	0.777
Potassium (mg/dL)	−0.3	(−0.7–0.1)	0.050	0.00	(−0.3–0.1)	0.403	0.255
HDL cholesterol (mg/dL)	−4.0	(−9.0–2.5)	0.248	2.0	(−4.0–4.5)	0.889	0.217
LDL cholesterol (mg/dL)	−5.4	(−12.5–−3.3)	0.016	2.0	(−9.0–3.2)	0.861	0.077
Cholesterol (mg/dL)	−9.0	(−23.5–5.5)	0.147	−7.0	(−18.0–5.5)	0.136	0.663
Triglycerides (mg/dL)	0.0	(−20.5–12.5)	0.563	−7.0	(−45.5–18.00)	0.345	0.644
Creatinine (mg/dL)	0.0	(−0.1–0.3)	0.442	−0.1	(−0.2–0.1)	0.327	0.293
Glucose (mg/dL)	2.0	(−17.5–23.5)	0.701	−2.0	(−13.0–27.0)	0.861	0.898
Glycated hemoglobin (%)	0.0	(−0.2–0.2)	0.632	0.0	(−0.4–0.1)	0.442	0.483
Hemoglobin (g/dL)	−0.1	(−0.6–0.6)	0.582	−0.2	(−0.7–0.2)	0.327	0.757
FGF 23 (ng/mL)	−0.2	(−12.6–23.0)	0.657	3.1	(−23.0–18.4)	0.861	0.762
Sodium (mEq/L)	0.0	(−1.0–1.5)	0.715	1.0	(−0.5–2.5)	0.118	0.253
Chlorine (mEq/L)	1.0	(−1.0–2.5)	0.287	−1.0	(−3.5–0.5)	0.199	0.067
Albumin (mg/dL)	−0.1	(−0.2–0.2)	0.391	−0.1	(−0.2–0.1)	0.258	0.958
Calcium (mg/dL)	0.0	(−0.4–0.2)	0.593	0.1	(−0.2–0.5)	0.478	0.354
Magnesium (mg/dL)	0.0	(−0.1–0.1))	0.538	0.1	(−0.1–0.2)	0.505	0.959
PTHi (pg/mL)	−2.0	(−14.1–10.0)	0.583	−3.5	(−15.0–3.25)	0.249	0.555
Alkaline phosphatase (u/L)	−6.0	(−8.0–2.5)	0.054	−2.0	(−9.5–13.0)	0.834	0.471
25-OH Vitamin D3 (ng/mL)	−0.6	(−3.9–2.0)	0.388	−3.2	(−5.0–1.7)	0.221	0.573
C-reactive protein (mg/L)	0.0	(−0.8–0.8)	0.929	0.1	(−0.7–2.0)	0.456	0.456
pH	−0.14	(−0.42–0.05)	0.133	−0.38	(−0.59–0.11)	0.045	0.425
Phosphate (mg/24 h)	−37.0	(−365–196)	0.650	9.0	(−76–169)	0.382	0.427
Potassium (mEq24 h)	−1.3	(−20.7–9.3)	0.173	0.7	(−4.5–17.5)	0.347	0.106
Tubular reabsorption (%)	−4.4	(−10.0–10.0)	0.937	−6.6	(−11.5–2.2)	0.041	0.268
Urea (mg/dL)	1.0	(−10.0–15.5)	0.551	4.0	(−8.5–13.0)	0.462	0.939
Glomerular F. (mL/min/1.73 m^2^)	−1.0	(−7.5–4.5)	0.479	2.0	(−4.15–5.50)	0.386	0.207
Microalbumin (mg/24 h)	−19.0	(−174.0–3.3)	0.011	26.0	(−59.7–30.5)	0.650	0.029
Uric Acid (mg/24 h)	0.0	(−0.6–0.3)	0.410	−0.1	(−0.6–0.1)	0.134	0.662
Protein (mg/24 h)	−10.0	(−183–7.5)	0.972	11.3	(−93.5–88.5)	0.600	0.101
Oxalate (mg/24 h)	−4.9	(−7.3–5.5)	0.279	0.3	(−14.2–4.5)	0.552	0.980
Sodium (mEq/24 h)	−21.0	(−58.3–4.5)	0.071	1.0	(−23.0–22.0)	0.917	0.096
Chlorine (mEq/24 h)	−26.4	(−68.0–5.5)	0.028	−11.0	(−54.9–27.5)	0.442	0.293

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
