# Peer review of "A Pilot Randomized Crossover Trial Assessing the Safety and Short-Term Effects of Walnut Consumption by Patients with Chronic Kidney Disease"

_nutrients, 2019, doi:10.3390/nu12010063_

Round 1
Reviewer 1 Report
Thank you for this interesting and well written manuscript. A randomized crossover trial looking into moderate CKD patients and dietary intake of walnuts and consequences on the phosphorus homeostasis and other cardio-vascular markers.
I have a few comments:
Materials and Methods
Were there any follow up during the study intervention by a dietician or physician? Was there a 24-h urine collection or only a urine sample? Moe et al (ref 39) looked into if a random urine phosphorus excretion could be an alternative to the currently recommended 24-hour urine collection. However, they did not find that fasting random urine collections offer such an alternative. How is it with Urinary InsPs excretion?
Figure 3
Change kalemia to potassium
Table 1: I do not understand table 1. The 13 patients included were either overweight or obese. Both 2000 and 1650 kcal/day sounds very little if the patients should be kept in steady state. Who were given which diet? How can you compare proteinuria if the patients were not given the same amount of protein?
Protein intake should not depend on kcal intake/day but on bodyweight according to guideline (Nutritional Management of Chronic Kidney Disease, NEJM) please change protein intake into g/kg/day Why were the fiber content not in accordance with the guidelines (25-30 g/day)?Author Response
Please see the attachment

Reviewer 2 Report
Thank you for the opportunity to comment on this paper. the study outlines the impact of 30g walnuts consumed daily for 4 weeks on a small cohort of people with CKD.
There are several areas that require clarification for the reader.
Please outline in more detail why 30g of walnuts was chosen and why 30 day intervention and washout periods Can you please describe in more detail the dietary plans. Why were 2000kcal and 1650 kcal chosen as standard given the varying requirements of people with stage 2 compared to stage 4 CKD. This is an important point to correct How was compliance to the intervention measured ? please explain Why was the significance level of 2 x the standard deviation considered significant Did you control for confounders like smoking in your analyses You have clearly expressed the primary and secondary outcomes but do not mention the other variables collected until the results section. Please outline clearly all variables collected in the methods section . for example there is no mention of urinary analyses such as microalbuminuria in the methods yet this is one of the key findings in the power calculations you discuss what these are but now they were derived. For example why did you choose to assume 95% would have no change in serum potassium ? in table 1 , i believe there is an error- i dont not see how you can achieve a 2000kcal per day diet with only 206 mg of sodium. the protein intake of these diets prescribed are also very low. you need weight in Table 2 to enable the reader to see how many g/kg of protein and potassium was prescribed in these diets as this may explain why you did not see elevations in the serum markers in Table 2 please report weight. This is important as it enables the reader to determine the potassium intake as mmol/kg which is standarrd for evidence based recommendations 10. the main findings of 19mg albumin, 4 mm SBP and 5.40 decline in LDL need ton placed into context. Tnhey may be statistically signfiancant but are they clinically signficant ? in the discissiojn the authors mention that they suggest a diet with a low inorganic P to protein ratio but do not report their own. Can this be supplied ? The authors also recommend blinded placebo controlled studies - how would you suggest this is conducted with a food based intervention ? I suggest reduce the strength of the conclusion. I do not believe the evidence is strong enough to suggest protection against cardiocavcualr events - perhaps reduction in biomarkers associated with CVDAuthor Response
Please see the attachment

Round 2
Reviewer 1 Report
None
Author Response
Dear reviewer,
Thank you for considering our paper entitled “A pilot randomized crossover trial assessing the safety and short-term effects of walnut consumption by patients with chronic kidney disease”. We thank both reviewers and the editor for their appreciation and constructive comments, which have led to a much improved version of our manuscript. We have made all the modifications that reviewer 2 suggest. We have attached the letters with the responses to each of the reviewer’s comments and the revised manuscript with highlighted changes.
We would deeply appreciate your consideration for the publication in Nutrients.
With my best regards,
Sincerely,
Pilar Sanchis, PhD

Reviewer 2 Report
Thank you to the authors for submitting their revision. I still have some concerns that I believe require amendment to ensure clarity for readers and ensure reproducibility.
You have indicated that 30g of walnut s was chosen as it is the amount per day to give you health befit but what health benefit ? fibre ? protein ? another nutrient – please specify as this is critical for our understanding Table 1 (amended) provides less information than before and believe it should be amended to include protein (g/kg/day), lipids g/day, CHO g/day, fibre (g/day), sodium (mg day), potassium mmol/kg and phosphorus per day. The amount of potassium in mmol per kg is important for readers especially dietitians to discern if the diets were low or high in potassium overall and provides insight into why there may or may not be change in serum markers such as K or PO4 Table 1 also needs all 4 groups and I am unclear why they were reduced eg 2000kcal and no nuts, 2000kcal with nuts, 1650 kcal with no nuts, 1650kcal with nuts Again I question your calculation – 206.9 mg of sodium in a 2000 kcal diet – is this really correct? Table 2 now contains addition of BMI but I would still like to see average weight The conclusion is still too strong – consumption of nuts appeared safe when included in a sodium, protein, phosphate and potassium controlled diet – not ad libitium - is key here
Author Response
Comment 1. You have indicated that 30g of walnut s was chosen as it is the amount per day to give you health benefit but what health benefit? fibre? protein? another nutrient – please specify as this is critical for our understanding
Response 1. Many thanks for this comment. Several health benefit have been described for a 30-40g of walnuts daily intake (1-1,5 oz). 30 g of walnuts contains 183 calories, 18g of fat, 0.6mg of sodium, 20 mg of calcium, 0,72 mg of iron, 3.8g of carbohydrates, 1.9g of fiber, 0.7g of sugars, 4.3g of protein and 480 mg of phytate (1,2). Walnuts are also a good source of vitamin E and vitamin B6 which have antioxidant properties (1).
Walnuts are a low carbohydrate option that contains plant-based omega-3 alpha-linolenic acid. Research has shown that diets rich in omega-3 fatty acids may reduce the risk of cardiovascular disease and may benefit those with type 2 diabetes, especially those with elevated triglycerides (3). In addition to being a healthy fat, walnuts are a good source of protein and fiber (1). Studies have shown that people who eat a high fiber diet are more likely to maintain a healthy weight and have a reduced risk of heart disease and cancer.
The monounsaturated and polyunsaturated fatty acids found in walnuts have been shown to decrease LDL cholesterol and triglyceride levels. For example, in a recent study in 194 healthy adults, eating walnuts daily for eight weeks produced a 5% decrease in total cholesterol, 5% decrease in LDL cholesterol and 5% decrease in triglycerides, compared to not eating walnuts (4).
A study published in the British Journal of Nutrition showed that the risk of coronary heart disease is 37 percent lower for those consuming nuts more than four times per week, compared to those who never or rarely consumed nuts (5).
The four-year PREDIMED study in about 7,500 adults at high risk of heart disease tested a Mediterranean diet supplemented with 1 ounce (28 grams) of mixed nuts daily, of which half were walnuts. At the end of the study, people on the nut-enriched Mediterranean diet had a 0.65 mmHg greater decrease in diastolic blood pressure than people on a similar heart-healthy control diet who weren’t given nuts (6).
Results of a meta-analysis published in 2009 suggested that a diet that is high in walnuts is linked to improved lipid and cholesterol profiles. The researchers also concluded that walnuts may also help reduce oxidative stress and inflammation (7).
Walnuts are a good source of the mineral copper, calcium and phytate and their low physiological levels have been associated with lower bone mineral density and an increased risk of osteoporosis (1,7). Furthermore, phytate prevents the formation of pathological calcifications, such as renal calculi (9,10), dental calculi (10), and cardiovascular calcification (11-13). Moreover, phytate may also provide protection against cancer (14) and Parkinson’s disease (15). In this sense, beneficial effects of consuming a walnut-rich diet in maintaining brain health with age have been indicated for several authors (16).
In 2003, the U.S. Food and Drug Administration (FDA) approved the claim for food labels on a variety of nuts, including walnuts, that: "Eating 1.5 ounces per day (40g aprox) of most nuts as part of a diet low in saturated fat and cholesterol may reduce the risk of heart disease."1. SELFNutritionData. Nuts, walnuts, English [includes USDA commodity food A259, A257] nutrition facts & calories. 2018.
Raygan F, Taghizadeh M, Mirhosseini N, et al. A comparison between the effects of flaxseed oil and fish oil supplementation on cardiovascular health in type 2 diabetic patients with coronary heart disease: A randomized, double-blinded, placebo-controlled trial. Phytother Res. 2019;33(7):1943-1951. doi:10.1002/ptr.6393 Prieto RM, Fiol M, Perello J, Estruch R, Ros E, Sanchis P, Grases F.Effects of Mediterranean diets with low and high proportions of phytate-rich foods on the urinary phytate excretion. Eur J Nutr. 2010 Sep;49(6):321-6. doi: 10.1007/s00394-009-0087-x. Epub 2010 Jan 28. Bamberger C, Rossmeier A, Lechner K, Wu L, Waldmann E, Stark RG, Altenhofer J, Henze K, Parhofer KG. A Walnut-Enriched Diet Reduces Lipids in Healthy Caucasian Subjects, Independent of Recommended Macronutrient Replacement and Time Point of Consumption: a Prospective, Randomized, Controlled Trial. Deirdre K Banel and Frank B Hu. Effects of walnut consumption on blood lipids and other cardiovascular risk factors: a meta-analysis and systematic review1,2,3Am J Clin Nutr. 2009 Jul; 90(1): 56–63. Toledo E, Hu FB, Estruch R, Buil-Cosiales P, Corella D, Salas-Salvadó J, Covas MI, Arós F, Gómez-Gracia E, Fiol M, Lapetra J, Serra-Majem L, Pinto X, Lamuela-Raventós RM, Saez G, Bulló M, Ruiz-Gutiérrez V, Ros E, Sorli JV, Martinez-Gonzalez MA. Effect of the Mediterranean diet on blood pressure in the PREDIMED trial: results from a randomized controlled trial. BMC Med. 2013 Sep 19;11:207. doi: 10.1186/1741-7015-11-207. Claire E. Berryman,5 Jessica A. Grieger,5 Sheila G. West,5,6 Chung-Yen O. Chen,7 Jeffrey B. Blumberg,7 George H. Rothblat,8 Sandhya Sankaranarayanan,8 and Penny M. Kris-Etherton5,*Acute Consumption of Walnuts and Walnut Components Differentially Affect Postprandial Lipemia, Endothelial Function, Oxidative Stress, and Cholesterol Efflux in Humans with Mild Hypercholesterolemia1,2,3,4 J Nutr. 2013 Jun; 143(6): 788–794. López-González, A.A., Grases, F., Monroy, N., Marí, B., Vicente-Herrero, M.T., Tur, F. & Perelló, J. Protective effect of myo-inositol hexaphosphate (phytate) on bone mass loss in postmenopausal women. Eur. J. Nutr. 52, 717-26 (2013). Grases, F. & Costa-Bauza, A. Phytate (IP6) is a powerful agent for preventing calcifications in biological fluids: usefulness in renal lithiasis treatment. Anticancer Res. 19, 3717–3722 (1999). Grases, F., Isern, B., Sanchis, P., Perello, J., Torres, J.J. & Costa-Bauza, A. Phytate acts as an inhibitor in formation of renal calculi. Front. Biosci. 12, 2580–2587 (2007). Grases, F., Perello, J., Sanchis, P., Isern, B., Prieto, R.M., Costa-Bauzá, A., Santiago, C., Ferragut, M.L. & Frontera, G. Anticalculus effect of a triclosan mouthwash containing phytate: a doubleblind, randomized, three-period crossover trial. J. Periodontal Res. 44, 616–621 (2009). Grases, F., Sanchis, P., Perello, J., Isern, B., Prieto, R.M., Fernandez-Palomeque, C., Saus, C. Phytate reduces age-related cardiovascular calcification. Front. Biosci. 13, 7115–7122 (2008). Sanchis, P., Buades, J.M., Berga, F., Gelabert, M.M., Molina, M., Íñigo, M.V., García, S., Gonzalez, J., Bernabeu, M.R., Costa-Bauzá, A. & Grases, F. Protective Effect of Myo-Inositol Hexaphosphate (Phytate) on Abdominal Aortic Calcification in Patients With Chronic Kidney Disease. J. Ren. Nutr. 26, 226-36 (2016). Fernández-Palomeque, C., Grau, A., Perelló, J., Sanchis, P., Isern, B., Prieto, R.M., Costa-Bauzá, A., Caldés, O.J., Bonnin, O., Garcia-Raja, A., Bethencourt, A. & Grases, F. Relationship between Urinary Level of Phytate and Valvular Calcification in an Elderly Population: A Cross-Sectional Study. PLoS One 10, 0136560 (2015). Vucenik, I. & Shamsuddin, A.M. Cancer inhibition by inositol hexaphosphate (IP6) and inositol: from laboratory to clinic. J. Nutr. 133, 3778–3784 (2003).15.Xu, Q., Kanthasamya, A.G. & Reddy, M.B. Neuroprotective effect of the natural iron chelator, phytic acid in a cell culture model of Parkinson's disease. Toxicology 245, 101-108 (2008).
Poulose SM, Miller MG, Shukitt-Hale B. Role of walnuts in maintaining brain health with age. J Nutr. 2014 Apr;144(4 Suppl):561S-566S. doi: 10.3945/jn.113.184838.
Comment 2. Table 1 (amended) provides less information than before and believe it should be amended to include protein (g/kg/day), lipids g/day, CHO g/day, fibre (g/day), sodium (mg day), potassium mmol/kg and phosphorus per day. The amount of potassium in mmol per kg is important for readers especially dietitians to discern if the diets were low or high in potassium overall and provides insight into why there may or may not be change in serum markers such as K or PO4.
Response 2. We thank the reviewer for this remark and we have changed the table 1 according his/her suggestions.
In this sense, we would like to clarify that our diets were based on the dietary recommendations of the kidney foundation for patients with CKD:
https://kidneyfoundation.cachefly.net/professionals/KDOQI/guidelines_bp/guide_6.htm
Comment 3. Table 1 also needs all 4 groups and I am unclear why they were reduced eg 2000kcal and no nuts, 2000kcal with nuts, 1650 kcal with no nuts, 1650kcal with nuts
Response 3. We thank the reviewer for this comment and we have added the four groups in table 1. Regarding this, initially we calculated the caloric requirements for each recruited patients. Based on the caloric requirements of our patients, we realized that the diets could be grouped into two groups in order to simplify the study: the 2000 kcal for men and 1650 kcal for women.
Comment 4. Again I question your calculation – 206.9 mg of sodium in a 2000 kcal diet – is this really correct?
Response 4. Yes, this is correct. The dietician elaborated the diets without added salt. For this reason and because the diet does not contain industrialized foods, the sodium content is minimal (200mg/day aprox). Nevertheless, patients had the option of adding a teaspoon of salt (5g of salt, 2000 mg of sodium) a day to their diet if they were unable to eat food without any salt. These diets follow the dietary recommendations for patients with CKD (< 2.4 g/day, table above). We have added this information in the legend of table 1.
Comment 5. Table 2 now contains addition of BMI but I would still like to see average weight
Response 5. Thanks for this remark and we apologize for this mistake. We have added the average weight to the table 2.
Comment 6. The conclusion is still too strong – consumption of nuts appeared safe when included in a sodium, protein, phosphate and potassium controlled diet – not ad libitium - is key here.
Response 6. Thanks for this remark we have changed the conclusion and added the sentence that the reviewer suggest in the abstract and in the conclusion.
